# Lateral Flow Immunoassays for Detecting Viral Infectious Antigens and Antibodies

**DOI:** 10.3390/mi13111901

**Published:** 2022-11-03

**Authors:** Rowa Y. Alhabbab

**Affiliations:** 1Vaccines and Immunotherapy Unit, King Fahad Medical Research Center, King Abdulaziz University, Jeddah 21589, Saudi Arabia; rymalhabbab@kau.edu.sa; 2Department of Medical Laboratory Sciences, Faculty of Applied Medical Sciences, King Abdulaziz University, Jeddah 21589, Saudi Arabia

**Keywords:** lateral flow immunoassay, viral infections, diagnostic tool, antigen detection, antibodies detection

## Abstract

Abundant immunological assays currently exist for detecting pathogens and identifying infected individuals, making detection of diseases at early stages integral to preventing their spread, together with the consequent emergence of global health crises. Lateral flow immunoassay (LFIA) is a test characterized by simplicity, low cost, and quick results. Furthermore, LFIA testing does not need well-trained individuals or laboratory settings. Therefore, it has been serving as an attractive tool that has been extensively used during the ongoing COVID-19 pandemic. Here, the LFIA strip’s available formats, reporter systems, components, and preparation are discussed. Moreover, this review provides an overview of the current LFIAs in detecting infectious viral antigens and humoral responses to viral infections.

## 1. Introduction

Infectious diseases, particularly viral infections, can cause asymptomatic to life-threatening illnesses and negatively impact the global medical system [1]. Viral infections can spread through communities via blood, blood products, sex, mosquitos, food, water, and aerosol. Viruses exist in various sizes ranging from 20 to 900 nm and typically consist of genetic materials that could be single or double-stranded deoxyribonucleic acid (DNA) or ribonucleic acid (RNA) enclosed by proteins, glycoproteins or lipid coating [2]. To survive, viruses require living cells suitable for hosting the virus replication process. In response to any viral infection, the immune system generates specific antibodies to neutralize the virus and the infected cells. However, the immune responses to viruses occasionally are not successful, due to rapid viral replication and spread within the host.

Over the years, humankind has endured several virus-mediated pandemics—from 1889 to 1890 with the Asiatic flu that destroyed around one million lives [3,4], followed by the Spanish flu (1918 to 1920), and then by the Asian flu from 1957 to 1968 [5,6]. Human immunodeficiency virus (HIV) has also been infecting people since 1981 and has caused the death of more than 33 million individuals [7]. Today, the newly emerged virus, severe acute respiratory syndrome coronavirus-2 (SARS-CoV-2), has resulted in a global outbreak of coronavirus disease (COVID-19) for more than two years, due to viral evolution. During the COVID-19 pandemic, many lives have been lost, partially due to inadequate testing, particularly in countries with insufficient resources. As a result, many undetected COVID-19-infected symptomatic or asymptomatic individuals were released to the community and were subsequently able to spread the infection silently in society [8]. At the beginning of the pandemic, real-time reverse transcriptase polymerase chain reaction (RT-PCR) testing was the standard method of detecting COVID-19-infected subjects; however, this technique has many limitations, including the costly testing procedure, the need for highly skilled staff, and the need for fully-equipped lab settings. Therefore, developing a detection method, such as lateral flow immunoassays (LFIA), that can overcome all these limitations is essential. LFIA is a point-of-care test that detects viral antigens and the humoral immune responses to viral infections. Although LFIA can distinctively be performed without lab or trained personnel, yielding relatively rapid results, antigen-detecting tests (AT) such as LFIAs are typically less sensitive than molecular methods [9,10,11]. However, AT methods used to detect SARS-CoV-2 antigens have shown comparable specificity levels to RT-PCR, specifically, around 99% [9,10]. This review describes the LFIA’s components, preparations, formats and the relevant literature addressing LFIA with respect to infectious diseases.

## 2. LFIA

Unlike many immunological assays, the LFIA is a rapid point-of-care test that provides results in a short time without the need for previous training or laboratory settings. These advantages have made LFIA devices an attractive tool for numerous applications beyond infectious agents’ detection, extending its usefulness to studying environmental sciences, drugs, and food, and several clinical analyses [12,13,14,15]. LFIA is a simple immunoassay that depends mainly on the accumulation of antibodies or antigens conjugated to reporter molecules on certain designated areas, known as the test or the control areas, depositing capture molecules enabling the detection of analyte-conjugate complexes on the membrane of the strip.

## 3. Components of LFIA Devices

As shown in Figure 1, LIFA devices are generally composed of several essential components, including a backing card, a sample pad, a conjugate pad, a membrane, and an absorbent pad, all of which are organized in a specific manner to ensure the capillary flow of the reactants across the membrane. Described in Figure 1, below, are the strip components mentioned above. 

### 3.1. Backing Card

The various components of the LFIA strips are fixed onto a backing card to provide the whole system with rigidity and ease of handling. Typically manufactured from a plastic polymer [16,17,18], the card is divided into adhesive layers to adhere to the four major components of the LFIA strips.

### 3.2. Sample Pad

The sample pad is found on the first part of the strip, where the sample is applied. Two main types of materials are suitable to be utilized as a sample pad: cellulose [19] and glass fibre [20]. However, the material used for this part of the strip should undergo testing to ensure that the applied sample is optimally flowing before reaching the next component. Notably, the sample’s medium plays a vital role in the selection process of the sample pad material, and both should undergo testing to ensure optimal results. For example, for saliva and blood, the most commonly used samples to detect infectious agents, different types of sample pads must be applied to the strip. The composition of the saliva frequently varies based on the individual, time of collection, and the food and drinks consumed prior to the collection of the sample. Therefore, it is recommended to pre-treat the sample pad with an optimizing buffer to normalize the sample and maintain the system’s pH to avoid non-specific interaction [21]. As for the blood, an external filter or integrated special filter sample pad membrane are suggested as essential for preventing the flow of unwanted molecules [22,23,24]. Generally, pre-treating the sample pad improves the sample flow rate and assay reproductivity [25].

### 3.3. Conjugate Pad

The conjugate pad is where antibodies or antigens conjugated to reporter molecules are placed. Glass fibre is the most commonly used material for this part. However, other components are also available, including polyester and cellulose [21]. The selected material will specify the amount of conjugate to be absorbed and the speed of the conjugate’s release into the system. Some LFIA devices omit this component; instead, the conjugate is added directly to the sample [26,27].

### 3.4. Nitrocellulose Membrane

Typically, nitrocellulose membranes (NCM) are used to develop LFIA strips; however, in some instances, customized cellulose membranes have been utilized [28]. The NCM is the most critical component of the strip, the component in which all reactions occur, and in which the results readout appears. The NCM contains two major areas: the test and the control lines. NCMs are available with various pore sizes to control the flow speed rate of the reactants throughout the system. The selection of the membrane pore size mainly depends on the analyte size and the sample type [29]. Most commercially available NCMs are defined by their capillary flow time. The capillary flow time is the time needed for the sample anterior to travel over the membrane, typically a distance of 4 cm, and is usually defined as s/4 cm. Generally, pore sizes range from 1 to 20 µm, and the conversion into capillary flow time is approximated. For instance, 8 µm and 6 µm would equal 135 s and 180 s, respectively [21,29]. The slow flow speed is associated with higher capillary flow time [21]. As shown in Table 1, NCM with a large pore size provides lower capillary flow time, while NCM with a small pore size is associated with a high capillary flow time and slower membrane [29]. Therefore, slower NCM would increase the reaction time between the components of the NCM and the flowing conjugates as well as the analytes, thus enhancing the test sensitivity, but might increase the risk of non-specific binding [21,30]. Therefore, selecting the proper pore size is crucial for obtaining sensitive and specific LFIA strips [21]. Additionally, the sample’s viscosity is an essential factor to consider upon selecting the pore size of the membrane, because viscous samples, such as saliva, run slower than do non-viscous specimens (Table 1). The membrane’s thickness, which can be measured by gauges, is an another important factor to consider; considered together with porosity, it allows the strip developer to predict the amount of liquid needed to fill the pores of the membrane [31]. Moreover, the membrane’s thickness must be compatible with the thickness of the other components in the strip and the housing cassette to evade over-compression upon assembling the device [31]. Notably, all the commercially available NCMs in the market contains surfactant to create hydrophilic membrane and assist in the protein’s binding to the membrane.

### 3.5. Adsorbent Pad

Adsorbent pads are cotton liners located at the second end of the strip. These pads adsorb the remaining reagents, clearing up the strip background by maintaining the capillary flow across the membrane.

## 4. LFIA Preparation

Several commercial LFIA devices can be used to detect a wide variety of infectious agents in patient samples, together with the humoral responses to these agents. Nonetheless, as shown in Figure 2, the preparation process is generally the same, from preparing and assembling the major components of the strip to cutting and formulating the cassette.

### 4.1. NCM Preparation

As mentioned earlier, the NCM contains the test and the control lines that must be stripped into the membrane, which requires the consideration of several essential factors. These factors include the concentration of the reagents, the dispensing rate, and the stripping buffer. To optimize results, each reagent’s concentration requires adjustment according to the specific assay of interest, ranging from 0.5 to 2 mg/mL. The test and the control lines must be dispensed into the membrane using the appropriate dispenser, which might be a contact or non-contact dispenser. The dispensing rate depends on the membrane’s pore size; for instance, a membrane with large pores needs a decreased dispensing rate to achieve a similar line width to that of membranes with smaller pores. Based on experience, the dispensing rate is usually between 0.5 to 1 µl/cm to obtain a line width of 1mm. Notably, the width of the lines correlates with the signal intensity and assay sensitivity [31,32]. Phosphate buffered saline (PBS) is suitable for most of the proteins; however, some are sensitive to the pH and salt concentration of the stripping buffer. Therefore, the selected buffer should be re-evaluated in the presence of non-specific binding or upon obtaining weak signals.

### 4.2. Conjugate Pad Preparation

The conjugate solution can be applied to the conjugate pad either by using an air jet dispenser—the most recommended method—or by immersing the glass fibre in the conjugate solution. Since the immersion process is usually performed manually, it would require further optimization, and it is inevitably associated with inconsistencies concerning the conjugate solution’s volume upon preparing different batches of conjugate pads, unlike the automated air-jet method [20,21]. Generally, the conjugate pad can be pre-treated with a buffer that would adjust the system’s pH based on the sample type. To avoid aggregation of the conjugates, the buffer should not hold high salt concentrations. Moreover, some amount of detergents, proteins and polymers can be added to the buffer to help release the conjugate. Also, adding blocking reagents to the buffer can eliminate the need to block the NC membrane [33,34,35]. Notably, selecting the proper conjugate pad pre-treating buffer requires testing several buffers’ components and ingredients. After adding the pre-treating buffer to the conjugate pad, the pad has to be dried at 37 °C for at least 3 h before applying the conjugate solution. Moreover, the conjugate solution typically needs to be in a sugar-containing buffer to maintain long-term stability after drying the pad and interacting with the specimen [21,29,36].

### 4.3. Sample Pad Preparation

Sample pad treatment depends on the type of the sample intended and is usually applied to maintain the variability associated with the sample’s pH, viscosity, and salt concentration. Treating the sample pad with an optimized buffer acts as a blocking agent that can normalize the sample’s pH and salt concentration to enhance the test’s consistency and performance. Moreover, it improves the specimen capillary flow over the strip. Particularly when using saliva as a sample, the sample pad treatment usually contains salts and surfactants that break down mucins and proteins, thus decreasing the sample’s viscosity and improving flow through the strip. However, buffers containing salts and surfactants in LFIA devices designated for whole-blood samples are strongly not recommended because they may haemolyze the red blood cells within the specimen, causing the passage of unwanted lysed cells to the strip.

### 4.4. Assembling the LFIA Components

LFIA components can be assembled by using a fully automated system to produce large numbers of strips. For small-scale production, laminating the components can be done manually or by using lamination machines to place and hold the materials on top of the backing card.

### 4.5. Cutting the Assembled LFIA Strips

Following the assembly of the LFIA components, the backing cards containing all the materials are cut into strips. The cutting sizes of the backing cards are based on the design of the specific test requirements, usually ranging between 3 to 6 mm in width. Automated cutters such as a guillotine are preferred, to maintain manufacturing accuracy and reproducibility. Notably, thin strips are cost-effective but could lead to less accuracy due to the edge effect. For instance, if the strip width is 3 mm, then 0.5 mm of the strip edge is exposed to unusual flow, representing 33% of the strip, but if the strip is 6 mm, then only 16% of the strip width is affected [32].

### 4.6. Assembling of the Cassette

Typically, the cassette is designed after optimizing the entire system to fit the length, width, and thickness needed for all the components. LFIA strips must be housed in a solid, well-designed cassette that can be easily handled, which is also reproducible and reliable. This part of the LFIA device is a crucial component that is usually overlooked, one which applies pressure on specific points of the strip to maintain optimal flow control while conserving the flow rate. Moreover, the cassette design must be optimized to avoid strip flooding.

## 5. Common LFIA Principles Used in Designing Strips for Detecting Infectious Agents and the Immune Responses against Them

### 5.1. Competitive Format

Competitive LFIAs detect low molecular weight analytes, such as drugs and toxins, that will not bind into two antibodies simultaneously [37]. The optimization of this format is straightforward and time-efficient compared to the other manufactured LFIA formats. Upon applying the sample to the sample pad, the sample components reach the antibody-reporter molecules placed in the conjugate pad. Suppose the target analyte is present in the sample; the antibody-reporter molecules conjugate will then recognize the analyte in the sample and develop a complex that, at the test line, will not bind to immobilized antigens similar to the sample’s target analyte (Figure 3). Therefore, the absence of color at the test line indicates a positive result, while the presence of color at the test line represents a negative result [38]. However, the signal intensity at the test line is inversely proportional to the analyte concentration of the sample, which may present difficulties when interpreting the results. In other words, a weak signal indicates that the analyte is still present in high concentration, while a strong signal implies that the analyte concentration is low or absent. The control line usually contains, e.g., goat anti-mouse antibodies, while the corresponding conjugate would be coated with mouse antibodies and placed within the conjugate pad; the test is invalid if the control line does not show a colorimetric signal. This test format can detect multiple targets simultaneously in a single sample, multiplex competitive LFIA [25].

### 5.2. Sandwich Format

Unlike the competitive format, the sandwich format is typically used to detect large molecules such as microorganisms and proteins. It provides more specificity by using two antibodies that recognize the same antigen at two distinct epitopes (Figure 4) [39,40]. However, finding pairs of antibodies with these criteria is challenging and time-consuming, especially when dealing with small analytes. Difficulty may also arise due to steric hindrance blockage between the two antibodies [41]. Therefore, this test format becomes less viable with increasingly small molecules [42].

In contrast to competitive LFIA devices, the presence of color at the test line indicates that the results are positive, but the absence of color represents negative results [43,44,45]. The most common use of this principle is in pregnancy tests, which detect the elevated level of the chorionic gonadotropin hormone that increases during pregnancy within a range between 10 and 25 U/mL based on the test sensitivity [46,47].

Generally, competitive and sandwich systems have specific advantages and drawbacks based on the type of analytes intended to detect, the analytes’ concentration, and the assay design [37]. For instance, the sandwich format retains high analytical sensitivity but may yield false negatives at high concentrations due to the “high dose effect”, which also known as “hook effect”. Conversely, the competitive format can detect analytes at high amounts but has a more limited sensitivity range than the sandwich format [38,44,46]. Table 2 shows the major differences between the two LFIA formats.

## 6. Type of LFIA Reporters

Several types of reporter, or labelling, molecules, such as colloidal gold nanoparticles, magnetic nanoparticles [48], fluorescent probes [18,49], enzymes [50], and many others, have been identified as being suitable for generating detectable signals upon developing LFIAs. The following sections discuss the systems most commonly used to develop LFIA strips for infectious agents, and Figure 5 illustrates the major types of reporter molecules.

### 6.1. Colloidal Gold Nanoparticles (AuNP)

This type of reporter is highly stable in its liquid and dried forms, and is frequently suspended in a water-based liquid. Typically, this solution comes in a scarlet suspension; however, it can also be a blue or purple liquid based on the particles’ size, according to the user’s preference. Notably, increasingly large-sized particles can cause aggregation of the gold nanoparticles, while increasingly smaller particles become more challenging to wash and may generate weak colorimetric signals. The most recommended particle size is 40 nm, which can provide the maximum color intensity with less steric hindrances of the antibody conjugates [27]. Colloidal gold nanoparticles (*Au*NP) are associated with many benefits, such as the simplicity of the conjugation procedure, the ability to manage the antibodies orientation on the modified surface of the gold nanoparticles [51], and a low cost compared to other reporters.

Moreover, *Au*NP is stable, and the obtained color on the LFIA strips can therefore last for longer durations. Colloidal gold nanoparticles are also simple to synthesize with the preferred surface functional group [52,53] and are commercially available from several sources. However, achieving the optimal concentration of antibodies to be used with colloidal gold nanoparticles is critical for successfully developing competitive LFIA strips. Applying low antibodies concentration necessitates to more sensitive assays; nevertheless, weak colorimetric signals would require the use of higher concentrations of the antibodies. Using *Au*NP is associated with minor disadvantages, such as possibly obtaining false-positive or negative results in response to excess salt or high pH effects. Furthermore, the pink color of the particles is not suitable upon examining samples of the same color.

### 6.2. Quantum Dots (QDs)

These nanoparticles are semiconductors, and their composition dictates their optoelectronic properties. The QD particles usually range from 1.5 to 10 nm [27]. Despite being costly, QDs possess multiple advantages, including their high colloidal and chemical stability and reduced photobleaching rate. Assays using QDs are also highly sensitive with visually detectable results; however, these results should be used as a final readout due to the sharp drop-off in the test sensitivity. QDs display more photostability, higher fluorescence, and broad absorption spectra with exclusive emission wavelength compared to organic dyes such as fluorescein and R-phycoerythrin [54]. While QDs are valuable in LFIA production, the production procedure is toxic to the environment and must follow proper waste disposal guidelines.

### 6.3. Magnetic Particles and Aggregates

Magnetic particles (e.g., Fe_3_O_4_ particles) have been used in several reports as colored reporters to develop LFIA [55,56,57]. However, the color generated by magnetic particles at the test line has to be measured by an optical reader. At the same time, a magnetic reader can record the magnetic signals produced, allowing them to be of further use as a detection signal. The magnetic signals produced by these particles are characterized by their long-lasting stability, particularly upon comparing them to the optical signals, and increase the LFIA sensitivity from 10 to 1000 folds [58]. A significant drawback of iron oxide nanoparticles is that their absorption spectrum that covers the entire visual area is drab. However, poly ethylene glycol can adjust magnetic iron oxide particles into different-sized aggregates via poly-L-Lysine cross-linking, making them more sensitive in their detection level than single iron oxide nanoparticles [59].

### 6.4. Fluorescent and Luminescent

Fluorescent molecules have been used extensively in the development of LFIA. The value of applying these molecules in LFIA is their ability to quantify the amount of target analyte in the tested sample. Organic fluorophores (e.g., rhodamine) have been exploited to develop protein-detecting LFIA strips [60,61]. Although optimally functional LFIA requires that organic fluorophores be highly stable and bright, these molecules are coupled with photobleaching issues that decreases the test sensitivity [62]. Gradually, QDs are replacing organic fluorescent dyes for several reasons, including the QDs’ unique optical and electrical properties.

### 6.5. Enzyme

Enzymes can also be used to develop LFIA to produce color or chemiluminescence [63,64]. However, a substrate must be applied to the LFIA strip in an additional step to obtain the results. The assay’s sensitivity is based mainly on the selected enzyme-substrate combination. Using enzyme-loaded gold nanoparticles has increased the sensitivity of the LFIA tests that use enzymes as a reporting system [65].

### 6.6. Surface-Enhanced Raman Scattering (SERS) Tags

The main components of the SERS tags include Au nanoparticles (plasmonic core), adsorbed or embedded Raman-active molecules as nanoparticles, and a layer of inorganic or protective polymer [66]. The optical efficiency of these tags is much higher than that of the fluorescent dyes, including organic dyes and QDs [67]. Physiochemically, these tags resemble simple colloidal gold nanoparticles (AuNP). Therefore, they can be applied in the same LFIA design. From the optical point of view, SERS tags have multiple advantages, such as the possibility of detecting even a single tag via a common optical technique [68]. The signal’s intensity obtained using SERS is proportional to the number of tags. Moreover, SERS tags are a suitable choice due to their narrow Raman scattering band spectra in developing multiplex LFIA.

## 7. LFIA for Detecting Viral Infections

### 7.1. COVID-19

SARS-CoV-2 emerged in 2019, driving the COVID-19 pandemic, which has lasted more than two years, due to the virus’s continued evolution. Many variants of SARS-CoV-2 have emerged, and some have been reported to possess higher transmission rates and could be more pathogenic than the wild-type virus [69,70,71,72]. Since COVID-19 symptoms are non-specific and can range in severity from asymptomatic to fatal, it becomes essential to have a reliable diagnostic system that would help detect early-stage infection; thereby assisting in isolating confirmed cases to contain the spread of the virus. Although RT-PCR is the golden standard for detecting viral RNA in samples obtained from infected individuals, it can produce false-negative test results. This discrepancy can be due to several reasons, including the quality and time of sample collection, since the viral load in the upper respiratory tract declines with time [73,74].

Moreover, RT-PCR is associated with several limitations, such as long turnaround time, high cost, the need for well-trained personnel, and the need for a fully equipped laboratory. Subsequently, many LFIA devices have been developed to circumvent these issues. Nevertheless, RT-PCR and LFIA devices detecting COVID-19 do not provide insight into the patient’s immune status. Therefore, the market has also introduced LFIA devices detecting humoral immune responses to SARS-CoV-2. Generally, virus-specific IgM antibodies in patient serum samples indicate active infection, while the existence of IgG alone implies that the patient has developed immunity to infection and mostly recovered. Hence, it might be critical to complement antigen and antibody detection assays to increase diagnostic accuracy and decrease the spread of any viral infection.

The first study to develop LFIA strips for detecting humoral immune responses, namely IgM and IgG antibodies, against SARS-CoV-2 infection in serum samples has conjugated *Au*NP to COVID-19 recombinant proteins, and also to rabbit antibodies. The two prepared conjugates were then mixed and sprayed onto the conjugate pad. Three lines were immobilized onto the assay NCM: a test line containing anti-human IgM antibodies, a second test line composed of anti-human IgG antibodies, and a control line with anti-rabbit IgG antibodies. In their study, the results were visualized in 15 min with a sensitivity of 88% and specificity of 90.6% [75]. Notably, as with all respiratory viral infections, the immune system produces IgM, IgG, and IgA antibodies [75,76]. For that reason, Cavalera et al. developed LFIA devices detecting the three antibodies in serum. The LFIA strips detected the total immunoglobulin, including IgM, IgG and IgA antibodies, against SARS-CoV-2-nucleocapsid protein (NP), through two test lines. The first test line of the assay contained protein A, which is known to bind to the IgG Fc region and, to some extent, to some IgM and IgA Fc domains. The second test line, in their setting, consisted of immobilized SARS-CoV-2-NP while spraying their control line with avidin. The conjugate pad used in this assay contained two conjugates: SARS-CoV-2-NP conjugated to gold nanoparticles as the test line reporter, and gold nanoparticles coated with biotin as the control line reporter. This test produced results within 20 min from placing the sample on the strip, with specificities and sensitivities of 100% and 94.6%, respectively [77].

Moreover, different approaches have been made to increase the sensitivity and specificity of the generated LFIA devices to detect humoral antibodies. For instance, gold nanoparticle signals were amplified by generating a complex of one gold nanoparticle conjugated to anti-SARS-CoV-2-NP IgG antibodies linked to five gold nanoparticles conjugated to SARS-CoV-2-NP. The target analyte in the serum samples reacted with the prepared complexes upon addition to the LFIA strips. The complexes streamed and captured with the immobilized anti-human IgM or IgG could then be pipetted onto the test line of the assay NCM. This test method requires no more than 2 µL from the sample to provide results, with an average sensitivity and specificity of 95% and 94.5%, respectively, within 15 min [78].

Extracting serum from whole blood requires well-trained personnel; generating LFIA devices that detect antibodies to COVID-19 directly from whole blood would be more reasonable. Therefore, Black and colleagues have produced LFIA strips that detect SARS-CoV-2 specific IgG and IgM antibodies in whole blood. Their test displayed high specificity and delivered results after 10 min [79]. Besides the conventional reporters, Lanthanide-doped polystyrene (LNP) nanoparticles have been used to develop LFIA strips to detect SARS-CoV-2 IgG antibodies in serum samples. In this case, after preparing the LNP nanoparticles, the LNP nanoparticles were conjugated to rabbit IgG and mouse anti-human IgG antibodies, then sprayed onto the strip conjugate pad. The test line of this assay contained recombinant SARS-CoV-2-NP, and the control line contained goat anti-rabbit IgG antibodies. However, the test sensitivity and specificity percentages were not calculated [35].

Furthermore, QDs have also been utilized to develop LFIA to detect humoral immune responses to COVID-19 in serum. QDs can assemble more biomolecules than conventional gold nanoparticles, thus increasing the strip’s sensitivity and accuracy. The QDs used in designing the COVID-19 antibody detecting LFIA were composed of a 20 nm monodisperse SiO_2_ NP hydrophilic core, a layer of 4 nm *Au*NP to generate strong colorimetric shell, and a layer of carboxylated QD to provide more surface sites to conjugate proteins and give high luminescence. The designed SiO_2_@*Au*NP@QD was then conjugated to SARS-CoV-2-spike protein (SP), and anti-human IgM and IgG antibodies were immobilized separately onto two NCM test lines. This technique demonstrated high sensitivity and specificity levels of 100% in detecting SARS-CoV-2 antibodies [80].

Several groups have developed LFIA to detect SARS-CoV-2 genomic materials; however, these LFIA usually require PCR or other techniques to amplify the detected nucleic acid [81,82,83]. Therefore, these techniques overpower the main advantages of using LFIA, namely that well-trained personnel, and a laboratory setting become unnecessary for rapid detection.

Global markets are becoming saturated with LFIA tests designed to detect SARS-CoV-2 antigens, particularly SARS-CoV-2-SP and NP. Diagnosing COVID-19 using SARS-CoV-2-SP, which covers the entire viral surface, can provide highly specific and sensitive LFIA devices [84]. Owing to this, several groups have generated LFIA to detect SARS-CoV-2-SP. For example, Liu and colleagues have applied the conventional sandwich immunoassay principle to generate their assay. Their study used CO-Fe@hemin peroxidase nano-enzyme, which can mirror the peroxidase activity, instead of the conventional nanoparticle reporters. These particles were synthesized through a hydrothermal step and applied as nano-enzyme labels that catalyzed the substrate, luminol/H_2_O_2_/enhancer and generated chemiluminescent signals. The prepared reporter was then conjugated to anti-SARS-CoV-2-SP antibodies that recognized the SARS-CoV-2-SP and was then captured on the NCM test line [85]. Despite the need to overcome the challenge of finding a matched pair of antibodies that can recognize the SARS-CoV-SP at different epitopes and produce a LFIA test with acceptable performance, not much has been reported on matching antibodies pairs against SARS-CoV-2 antigens for immunoassays application. In response, a rapid system was proposed by Lee et al. In their study, the SARS-CoV-2 entry receptor, ACE2, was immobilized onto the LFIA strip as the test line, and commercial antibodies specific to SARS-CoV-2-S1 protein were conjugated to red cellulose nanobeads. Their results are of great importance because it is the first to report the possibility of matching ACE2 and antibodies to detect SARS-CoV-2 antigens by LFIA [86].

Currently, several challenges of managing and diagnosing COVID-19 infection are increasing due to the emergence of multiple variants with various transmission rates and severity [87,88]. Most emerging variants have many mutations, mainly in the spike gene, with minimal alteration in the N protein [89,90,91,92]. Moreover, the virus’s N-proteins are the most abundant and released in large amounts, in almost all samples. In response, several studies have recently developed SARS-CoV-2-NP-detecting LFIA to ensure enhanced specificity and sensitivity when testing to detect the latest variants. Most of the reported LFIA strips to detect SARS-CoV-2-NP are based on the principle of the double antibodies sandwich system [93]. For instance, Zhang et al. have combined two technologies: the fluorescent microsphere and the immunochromatographic method. In their study, the first monoclonal anti-SARS-CoV-2-NP antibodies were immobilized onto the test line of the assay strip, while the second monoclonal anti-SARSA-CoV-2-NP antibodies were conjugated to the fluorescent microsphere, thereby developing LFIA strips that can detect recombinant N proteins with a limit of detection (LOD) of 100 ng/mL and 1 × 10^3^ TCID_50_/_mL_ for the active virus [94]. Miyakawa’s group has developed LFIA to detect SARS-CoV-2-NP by applying a silver amplification system based on the *Au*NP LFIA test. According to their design, the prepared strips with their amplification method gave higher diagnostic sensitivity levels than did the unamplified LFIA strips [95]. As COVID-19 spreads and mutates, it is essential to validate and evaluate LFIA devices to detect SARS-CoV-2 antigens. Continuous validation of these devices would help overcome the accuracy and reliability challenges associated with antigen rapid diagnostic testing due to the presence of several variants.

### 7.2. Influenza Viruses

Several LFIA devices have been generated to detect seasonal influenza viruses, classified into A, B, and C types [96]. Influenza type A is categorized into subtypes according to their surface proteins, such as neuraminidase (N protein) and hemagglutinin (H protein) [97]. Since influenza type A is highly infectious and can be associated with a severe respiratory infection, it is critical to detect infected individuals at the early stages by using a rapid method to avoid pandemics [98]. Therefore, many LFIA tests have been released to detect influenza type A. For example, Peng and colleagues developed the LFIA test to detect the H9 subtype of avian influenza virus using the classical *Au*NP conjugation process. The system applied two distinct monoclonal antibodies: 4C4 antibodies conjugated to the *Au*NP nanoparticles, and 4D4 antibodies stripped on the test line. The results were obtained within ten minutes, and the assay was highly specific and sensitive [99]. Using fluorescent (cy5) dropped-silica nanoparticles as a reporter system to detect the influenza virus has been used by few researchers. The comparative advantage of utilizing Cy5 dyes is the low background noise generated upon using them in LFIA strips. This dye has been reported to be more sensitive and photostable than organic fluorophores [100]. LFIA using this dye as a reporter showed eight-times higher sensitivity levels than the conventional LFIA using gold nanoparticles [101]. The newly introduced method SERS has also been used to improve the sensitivity of the conventional LFIA test.

The technique principle is similar to conventional LFIA strips, but uses different detection nanoparticles [102,103]. In SERS, molecules are detected and amplified by Roman scattering upon being absorbed on a rough metal surface, and the SERS nanotags are modified with *Au*NP. Typically, SERS nanotags are composed of three components: charged Roman dye molecules, gold or silver core-based metals, and surface-modified antibodies [104,105]. Magnetic SERS-tags make up another system used in 2019 to develop LFIA strips to detect human adenovirus (HAdv) and influenza type A H1N1 viruses. Fe_3_O_4_@*Au*NP (150 nm) was used in this method as the material for magnetic SERS nanotags. However, this technique requires an external magnetic field to separate the target analyte from the rest of the sample. The test does not contain a conjugate pad and consists of two test lines, H1N1 and HAdv, and a control line [106].

### 7.3. Human Immune Deficiency Virus

Human immunodeficiency virus (HIV) is another critical viral infection that can be spread sexually or via blood contact, including blood transfusion. HIV is a life-threatening viral infection, particularly upon advancing into acquired immune deficiency syndrome (AIDS) and continues to disrupt social life and exacerbate within areas of insufficient resources [107,108]. Therefore, LFIA detecting HIV with high sensitivity and specificity would significantly help, particularly in regions with limited resources. Hence, SERS-LFIA technology has been used to develop a highly sensitive test to detect and quantify HIV-1 DNA. The latter study used malachite green isothiocyanate (MGITC) as a Raman reporter with *Au*NPs gold nanoparticles plus detection DNA probes. A quantitative result can be determined by recording the change in the Raman intensity on the test line. This system has been reported to be 1000 times more sensitive than the other methods, including the colorimetric and the fluorescent systems [103]. Quantitative analysis using LFIA is a limitation that is overcome when applying SERS-based LFIA assays to quantify HIV-1 DNA. Rohrman et al. have also used an additional system to detect and quantify HIV. In their study, gold nanoparticles were conjugated to complementary oligonucleotides that bound to HIV-1 RNA and were captured by the target sequence in the test area. After washing the strip, applying an enhancement solution increased the optical absorbance of the captured gold nanoparticle conjugate. This test used only 20 μL from the tested sample and showed results within 20 min with 0.5 log copies/mL resolution [44]. QD is another system that has been introduced to generate LFIA for HIV. Deng and colleagues combined DNA with QD through a one-step hydrothermal procedure to detect HIV DNA in serum. The system detection limit ranged from 1 pM to 10 nM [109]. Some studies have also designed LFIA to detect HIV-1 P24 antigen by using QD [110,111]. For example, a multiplex fluorescent LFIA to detect several infectious agents, to include AIDS, hepatitis B virus (HBV), hepatitis C virus (HCV), and treponema pallidum antibodies, has been developed by using fluorescent probes containing QD [112].

### 7.4. Hepatitis and Ebola

The development of LFIA devices was not confined to detecting HIV and the other aforementioned infectious viruses, but extended as well to hepatitis. The most common hepatitis types are hepatitis A, B and C, with type B as well as type C being transmitted mainly via exposure to blood from an infected individual; however, hepatitis A virus (HAV) is mainly transmitted through contaminated water or food [113,114,115]. Quantum dot-beads (QBs) of high luminescence were used to develop LFIA to detect the serum hepatitis B surface antigen (HBsAg). In this design, the conjugate pad contained antibodies against HBsAg coupled to QB, while the test line consisted of goat anti-HBsAg specific antibodies, and the control line had donkey anti-mouse polyclonal antibodies. The fluorescent signal generated at the test area and the control lines were captured by a fluorescent strip reader providing quantitative assessment. The assay showed excellent and quick results in 5 min, with a sensitivity of 75 pg/mL [116].

Moreover, LFIA to detect and determine HBV genotypes (A, B, C and D) were developed, using smartphones as tools to measure the generated fluorescent signal [117]. Multiplexing to detect HBV genotypes has also been reported by Sung et al., where they used in their system well-controlled *Au*NP with restricted size distribution [118]. Multiplex LFIA devices to detect HCV and HIV have also been developed using seven different proteinticle beads as 3D probes that exhibited various viral antigens on their surface. The method has been reported to be more sensitive than those using peptide probes, which were estimated to have 65 to 90% diagnostic sensitivity. Therefore, this technique to develop LFIA is meant to overcome the associated issues with other probes that could cause clustering, inactivation, irregular flow, and assay surface instability, thus lowering the test accuracy [119]. Ebola virus infection is associated with life-threatening bleeding, and 50 to 90% of the infected cases do not survive. A LFIA test detecting Ebola glycoproteins was developed with a reporter of nanospheres that consisted of hundreds of QDs and dozens of *Au*NP nanoparticles. This reporter system was simultaneously modified with the antibodies and streptavidin to bind to the Ebola glycoproteins and enhance the signals [120].

### 7.5. Dengue and Zika

Mosquitoes serve as mobile vehicles transmitting several infectious viruses to humans, including dengue and Zika [121,122]. Since the same mosquito species transmit both viruses, those two viruses have a close relationship. Although recovering from dengue results in lifelong immunity, the developed antibodies can facilitate the entry of different viral serotypes, such as Zika, into the cells, causing more severe infection [123]. Therefore, it is essential to have rapid, reliable and cheap LFIA to detect the two infections simultaneously. Yrad and colleagues developed LFIA to detect dengue-1 RNA using dextrin-capped *Au*NP nanoparticles. Their study used the nucleic acid sandwich assay principle, utilizing a biosensor of three DNA probes. The three DNA probes consist of *Au*NP-labelled reporter probes, a probe specific to capture DENV-1 dDNA located at the test line, and a control probe placed on the control area. This system was designed to detect negative sense target RNA that can be amplified at 41 °C isothermal reactions without requiring thermocycling from nucleic acid sequence-based amplification (NASBA) [124]. Roman system has also been used to develop LFIA that can distinguish between dengue and Zika biomarkers, non-structural protein 1 (NS1). The marker probe was a gold nanostar conjugated to antibodies specific to Zika or dengue, while monoclonal antibodies specific to Zika and Dengue NS1 protein were immobilized on the LFIA test line. The study reported that using SERS to generate LFIA strips to detect dengue and Zika can produce a highly sensitive assay, thus improving the disease diagnosis at early stages even when the biomarker levels are low [125]. LFIA assays detecting dengue and Zika are developed mainly for use in tropical regions, where the viruses are most dominant. Therefore, Jeon et al. have developed SERS-based LFIA to be stable and reproducible at the temperatures of these areas. The group has used silica-encapsulated *Au*NP to overcome the thermal issue associated with the conventional LFIA, and the prepared strips using this system showed similar LOD despite the increase in temperatures. Therefore, this method helps detect dengue and Zika and could also be used for any viruses found in tropical reservoirs [126]. An additional study has documented a system using magnetic nanoparticles that can either be detected by a magnetic reader for quantification or by adding amplifying steps to detect the results visually. This method has combined magnetic nanoparticles with horseradish peroxidase (HRP) as an amplification system for visual detection. The system detected Dengue NS1 proteins by coating biotinylated magnetic nanoparticles with specific antibodies for Dengue NS1 protein. The absence of visually visible results at the test line, which contains capture monoclonal antibodies specific for Dengue NS1 protein, requires the addition of streptavidin-poly-HRP conjugate onto the strip. After washing, the color will appear upon the addition of the substrate. This system’s advantage is that the associated detection limit was 0.25 ng/mL for DENV-1 and DENV-3, 0.1 ng/mL for DENV-2, and 1 ng/mL for DENV-4, which is approximately 50-fold higher than the previously reported methods [127]. Recently, smartphones have been introduced as a tool to develop fluorescent LFIA platforms to detect Zika NS1 proteins. The smartphone in such a system would have a 3D printed plug-in fit for the phone’s external optical and electrical parts. The fluorescent LFIA strips used with this device utilized QD microsphere probes due to their high fluorescent signals. The fluorescent intensity of the captured QD by the antibodies placed at the strip, test and control lines are excited by the plug UV light and then detected by the smartphone camera. The detection levels associated with this approach are significantly high, 0.045 ng/mL [128]. Table 3 summarizes most of the studies mentioned in this review.

## 8. Challenges

Although LFIA devices have several attractive advantages, some of the LFIA tests are not sensitive enough for clinical applications. Therefore, many engineering and biochemical challenges must be resolved to produce LFIA devices up to the required level of sensitivity for clinical use. The inconsistency associated with paper-based materials and the variations between batches due to the different fabrication processes between manufacturers has to be addressed [129]. To overcome these variations between manufacturers in the capillary flow rate, pore size distribution, permeability, surface area, and wettability, the membranes must be treated physically and chemically [130]. Membrane treatment with surfactant and chemicals improves the membrane wettability but can reduce the capacity of the protein to bind to the membrane. Thus, manufacturers must use the proper optimized detergent concentrations during the fabrication process of the membrane. Charbaji and colleagues have introduced a cost-effective process to produce cellulose fibre embedded with zinc microparticles, known as zinculose, through sedimentation and without using chemicals to be used as a membrane in paper-based devices [131]. An additional effort to solve the irreproducibility of the flow on top of NCM is through introducing an alternative of adjacent repetitive vertical pillars known as pillar forest. This repetitive pattern creates continuous liquid streams into them [132]. Therefore, it provides higher flow reproducibility because of the controlled structure of the material. The pillar forest has been fabricated of silicon with polymer materials such as cyclo olefin polymer, polymethylmethacrylate or dextran [133,134]. Separating plasma from whole blood is an obstacle that has been solved by adding a plasma filter into the strips. However, protein retention in these filters reduces their performance. Guo et al. introduced synthetic papers to separate plasma by blood cells agglutination and capillary separation of plasma, and they recovered at least 82% of the plasma and the proteins in the sample [135]. However, using innovative materials requires validation for performance to meet the market’s level of acceptance.

Advances in printing enhanced the process of depositing a wide range of materials on the different parts of the assay, thus improving testing performance and reproducibility [136]. For example, it has been reported that inkjet printing of protein and enzymic solution on paper can occur without losing any of the properties of these molecules [137,138]. Moreover, it has been shown that dispensing the test lines in a line instead of a circle can reduce the drying effect and coffee-ring-effect, thereby improving the test performance [34].

It is inaccurate to quantify the colorimetric results obtained from LFIA devices by the naked eye. Therefore, the reader needs to digitalize the data into numbers [139]. Currently, smartphone cameras have been used to obtain images and immediate analysis [136,140,141]. Notably, the image quality may vary due to the amount of light or the type of software and hardware used during the process [139]. Several strategies have been employed to resolve the issues affecting image quality, and thus the development of smartphone sensors for paper-based assays has rapidly increased in the last years [142].

The assay reagents’ stability can influence the resulting signals [143]. Enzymes, for example, are applied in LFIA to amplify the obtained signals; however, many factors including, temperature, pH, and humidity can affect the activity of the enzymes used in the test [139,144]. In response, sugar is used to stabilize the enzyme contained in the assay [144,145]. Also, the fluorescence and other types of reporters have been applied in developing LFIA assays to amplify the results signals; however, most reporters need an additional reader with certain specifications to measure the results.

## 9. Conclusions and Future Prospects

This review has discussed the various LFIA components, the preparation of each component, the principle of the most common LFIA formats, the different types of reporter systems, and the various LFIA methods to diagnose COVID-19, influenza, HIV, hepatitis, Ebola, dengue and Zika. Infectious diseases have caused many epidemics and pandemics worldwide, with COVID-19 as a current example. LFIA is an excellent device that can aid in controlling the spread of highly infectious agents such as COVID-19 due to its low cost and the ability to obtain results within minutes without the need for laboratory settings or extensive training. However, the LFIA system has many limitations that must be addressed to facilitate the test performance; several technologies, such as SERS-tags and catalytic enhancements, have been reported to enhance the sensitivity of the LFIA system.

Nevertheless, developing new labels with more stability and efficiency is needed along with more interdisciplinary studies on detection methods. Moreover, portable and electronic devices to quantify the obtained results from certain LFIA devices should be reduced in size and made more accessible to users. In summary, LFIA devices are essential tools, particularly during epidemics and pandemics; however, the associated limitations need continuous and quick solutions.

## Figures and Tables

**Figure 1 micromachines-13-01901-f001:**
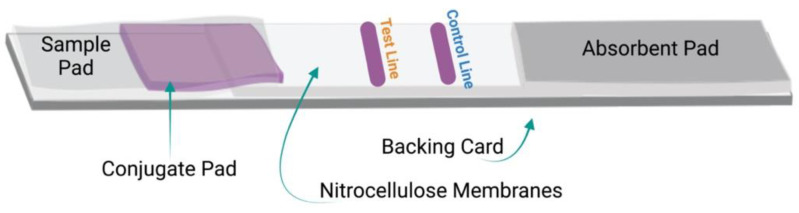
Components of LFIA devices: Schematic of the general components and design of the LFIA strips.

**Figure 2 micromachines-13-01901-f002:**
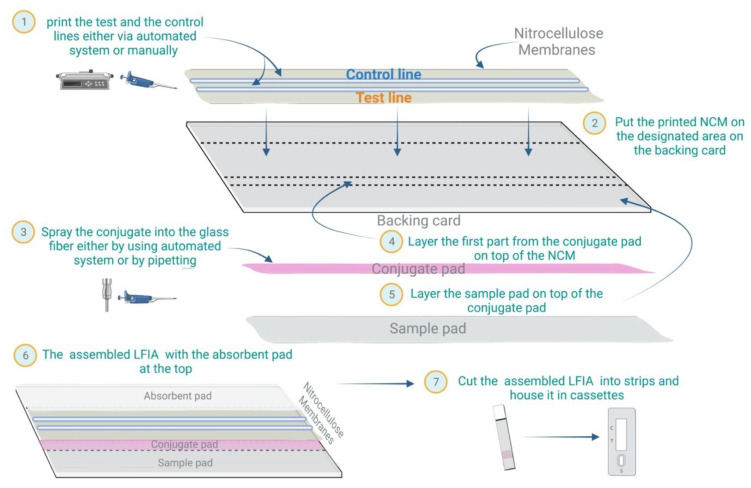
LFIA preparation: Schematic of the general steps for preparing LFIA devices.

**Figure 3 micromachines-13-01901-f003:**
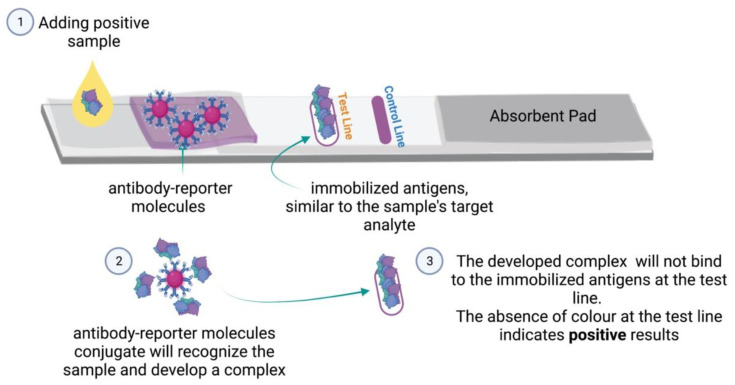
**LFIA competitive format:** Schematic of the competitive format LFIA components and principles.

**Figure 4 micromachines-13-01901-f004:**
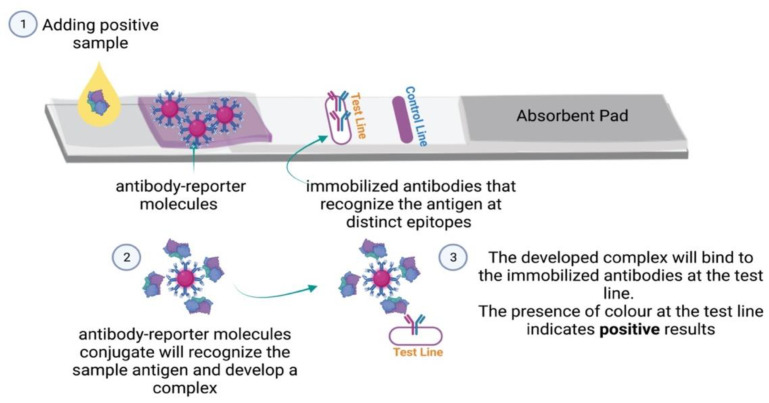
LFIA sandwich format: Schematic of the competitive format LFIA components and principles.

**Figure 5 micromachines-13-01901-f005:**
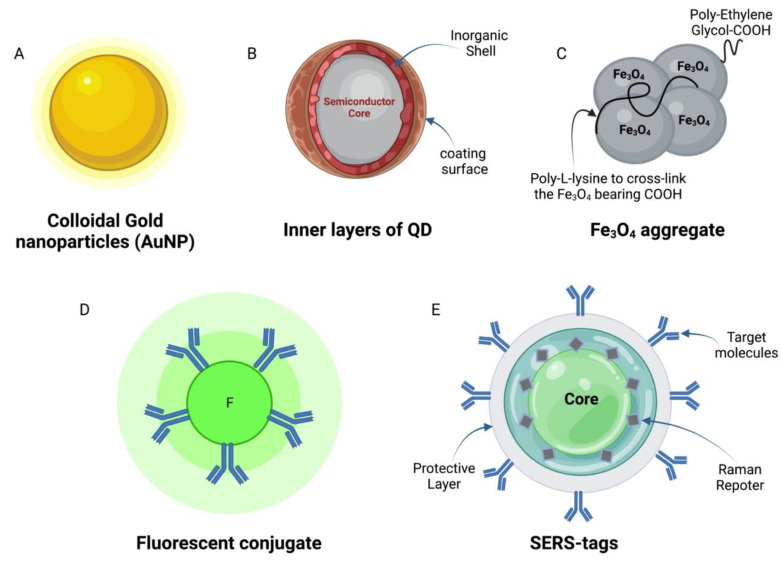
**Types of LFIA reporters:** (**A**) Colloidal gold nanoparticles (AuNP), (**B**) the inner layers of the QDs, (**C**) an example of the magnetic particles (Fe3O4) aggregate, (**D**) fluorescent conjugate, and (**E**) SERS-tag.

**Table 1 micromachines-13-01901-t001:** Relationship between the sample type, the NCM pore size, and the NCM flow speed.

Sample Type	Recommended Pore Size	Flow Speed	Examples of NCM Capillary Flow Time
Viscous, such as saliva	Large	Fast	75 s and 80 s
	Medium	Medium	120 S and 135 s
Non-viscous, such as urine	Small	Slow	170 s and 180 s
	Medium	Medium	120 S and 135 s

**Table 2 micromachines-13-01901-t002:** Competitive versus sandwich LFIA.

	Competitive Format	Sandwich Format
Type of analytes	Low molecular weight analytes	Detect large molecules
Positive results readout	No signals	Signals
Negative results readout	Strong signals	No signals
Examples	Drugs and toxins	Microorganisms and proteins

**Table 3 micromachines-13-01901-t003:** List of some of the LFIA based studies for detecting infectious diseases.

Disease	Reporter	Target	LFIA Performance	Sample	Time	Ref
COVID-19	Gold nanoparticles	IgM and IgG	Overall testing sensitivity and specificity 88% and 90.6%, respectively	serum	15 min	[75]
COVID-19	Gold nanoparticles	IgM, IgG and IgA	Total antibodies response to infection specificity and sensitivity 100% and 94.6%, respectively	serum	20 min	[77]
COVID-19	Gold nanoparticles	IgM and IgG	Sensitivity IgM 94%, sensitivity IgG 96%, specificity IgM 93%, specificity IgG 96%.	serum	15 min	[78]
COVID-19	Lanthanide-doped polystyrene (LNP) nanoparticles	IgG antibodies	*At*/*Ac* ratio cut-off value 0.0666	serum	10 min	[35]
COVID-19	CO-Fe@hemin peroxidase nano-enzyme	SARS-CoV-2-S Protein	Limit of Detection (LOD) for recombinant SARS-CoV-2-S Protein 0.1 ng/mL	-	16 min	[85]
COVID-19	Red cellulose nanobeads (CNB)	SARS-CoV-2-S1	LOD 1.86 × 10^5^ copies/mL	swab	20 min	[86]
COVID-19	Fluorescent microsphere in conjunction with the immunochromatographic method	SARS-CoV-2-N Protein	LOD for recombinant NP 100 ng/mL Activated SARS -CoV-2 virus 1 × 10^3^ TCID_50_/ml	swab	15 min	[94]
Influenza type A	*Au*NP	H9 subtype of avian influenza viruses (H9AIVs)	Sensitivity 0.25 units for H9AIV hemagglutinin (HA)	swab	10 min	[99]
Influenza type A	Fluorescent (cy5) dropped silica nanoparticles	Nucleoprotein of influenza A virion	LOD for recombinant nucleoprotein 250 ng/mL	-	30 min	[100]
HIV-1	MGITC@*Au*NPs	HIV-1 DNA	Detection limit 0.24 pg/mL	-	15 min	[103]
HIV-1	Quantum dot (QD)	HIV DNA	Detection limit 0.76 pM Detection range 1 pM to 10 nM	serum	15 min	[109]
Hepatitis B	Quantum dot-beads (QBs)	Hepatitis B surface antigen (HBsAg)	Sensitivity of 75 pg/ml	serum	5 min	[116]
HCV and HIV	Seven different proteinticle beads 3D probes	Structural and non-structural proteins of HIV and HCV	Sensitivity and specificity of 100%	serum	30 min	[119]
Ebola	QDs@*Au*NP	Ebola glycoproteins	Quantitative detection limit 0.18 ng/mL	Urine, plasma and water	20 min	[120]
Dengue	Dextrin-capped *Au*NP nanoparticles	Dengue-1 RNA	Cut-off value 1.2 × 10^4^ pfu/mL	serum	20 min	[124]
Dengue and Zika	SERS	Dengue and Zika, non-structural protein 1 (NS1)	LOD for Zika 0.72 ng/mL Dengue 7.67 ng/mL	serum	20 min	[125]
Dengue	Magnetic nanoparticles with horseradish peroxidase (HRP)	Dengue NS1 proteins	LOD 0.25 ng/mL for DENV-1 and DENV-3, 0.1 ng/mL for DENV-2, and 1 ng/mL for DENV-4	serum	25 min	[127]
Zika	QD microsphere probes	Zika NS1 proteins	LOD 0.045 ng/ml	serum	20 min	[128]

## Data Availability

Not applicable.

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
