# Peer review of "Lateral Flow Immunoassays for Detecting Viral Infectious Antigens and Antibodies"

_micromachines, 2022, doi:10.3390/mi13111901_

Round 1

Reviewer 1 Report

The manuscript by Alhabbab provides a review of lateral flow immunoassays for detecting viral infectious antigens and antibodies. Although the author did a good job in reviewing and including many interesting technologies used in creating LFIAs as well as various applications in which these devices are used, and while the report is of interest to the readers of Micromachines, the report would require a major revision before acceptance. I have several comments as per below:

The author is encouraged to proofread their manuscript. There are very few grammatical mistakes that should be avoided. In the abstract, “Making it an attractive tool that was extensively used during coronavirus disease 2019 (COVID-19) pandemic.” The pandemic is still ongoing and these lateral flow strips are commercially available and are still being used.

The author needs to clarify what they meant by “vectors” in the first paragraph.  

The author needs to clarify what they meant by their statement “and the subsequent release of COVID-19 infected symptomatic or asymptomatic individuals”.

The author needs to supplement the following statement with the drawback, mainly accuracy, of using LFIAs compared to PCR testing.  “Distinctively, LFIAs can be performed without the availability of lab or trained personnel while yielding relatively rapid results.”

The author needs to provide examples or references for the following statement “extending its usefulness to studying environmental sciences, drugs, food and several clinical analyses”

I believe the author meant to say “described in figure 1” instead of “Described in further detail below are the strip components mentioned above”

The author would need to upload a higher resolution schematic for figure 1 as the text is pixelated especially the test and control lines. The same is true for the remaining figures in the text.

What’s the difference between cellulose and cotton in “sample pad, including cellulose [12], cotton, and occasionally glass fibre [13].”

The author would need to clarify how the following can be done or provide a reference: “Therefore, it is recommended to pretreat the sample pad with an optimizing buffer to normalize the sample and maintain the system pH to avoid non-specific interaction.”

Does the author mean a sample pad that also acts as a filter in their statement “a filtered sample pad”?

It’s recommended, if possible, for the author to provide a reference or example for the use of these materials (i.e. polyester and cellulose) in the conjugate pad: “However, other components are also available, including polyester and cellulose.”

The author would need to provide more details in text on the selection process of the Nitrocellulose membrane. Is this selection based on a trial and error basis, based on prior experience or is there literature available to guide in the selection process? Is there any graph, chart or table for selecting the pore size as a function of the sample type such as blood or saliva? How would a researcher know how to choose what material, the film thickness for the same material, type or pore size of the nitrocellulose for their specific application? Does some of these films come with surfactant preloaded? 

The author needs to provide references for the following 2 statements: “The dispensing rate is usually between 0.5 to 1 μl/cm to obtain a line width of 1 mm. Notably, the width of lines correlates with the signal intensity and assay sensitivity.”

What does the author mean by “pleased” in “Most of the proteins are pleased with Phosphate Buffered Saline (PBS)”?

The following statement is not very clear, why would there be inconsistencies in the concentration if this is a flood coating process? “inconsistencies concerning the conjugate solution's volume upon preparing different batches of conjugate pads.” 

How and with what is the conjugate pad pre-treated in “the conjugate pad is pre-treated and dried before applying the conjugate solution.”

Is that the only reason why they use sugar in “sugar-containing buffer to maintain long-term stability after drying the pad and interacting with the specimen.” The author would need to clarify or add a reference to this statement. I was under the impression that sugar was added to reduce the rate at which the conjugate gets released as the sample flows since they don’t want it all to be washed away with the flow of the initial fluid front. 

The author needs to clarify what they mean by the edge effect in “thin strips are cost-effective but could lead to less accuracy due to the edge effect.”

The following sentence can’t stand on its own: “Suppose the target analyte is present in the sample.”

What’s on the control line in Figure 3? What does it capture and how would you know if the test is valid?

How would the results be interpreted if there’s a faint line on the test line? Is it a negative or a positive result in “However, the signal intensity at the test line is inversely proportional to the analyte concentration of the sample, which may present difficulties when interpreting the results.”

In what cases are competitive LFIAs vs. sandwich LFIAs used? This should be made clearer in the text.

The format of the references should be consistent with the rest of the manuscript and according to the requirement of the Micromachines journal in “presence of colour at the test line indicates that the results are positive (16-18)”

The paragraph that starts with “Several LFIA devices” and ends with “sensitivity of the conventional LFIA test.” In section 8 seems to have a different font than the rest of the manuscript.

Section 8 needs to have subsections for each of the viral infections discussed. It’s a very abrupt jump after discussing Covid in 9 paragraphs to discuss the seasonal influenza virus in the next paragraph.

Sections 7 and 8 take up 6 pages of text without a single figure that shows the different points being discussed. The author is required to add figures or tables wherever possible in these 2 sections to make it easier on the reader to follow up with what is being presented. 

The article finishes rather abruptly without a concluding section. The author is also required to add a “challenges and future trends” section before the conclusion section that summarizes the challenges faced using lateral flow strip technology in detecting viral antigens and antibodies and what the future trend going forward will be. The author can take note of recent advances in microfluidic technology such the use of composite material in lateral flow devices [1], the use of 3D printed lightbox and cellphone technology [2][3] to enhance the detection limit of these devices since that seems to be a critical issue in competitive LFIAs as the author has mentioned in text, the use of valve technology [4], optimized material selection, device architecture and sample processing to reduce the “coffee ring effect” or the influence of variable temperature and humidity conditions on device performance, etc.

[1] Charbaji, A.; Smith, W.; Anagnostopoulos, C.; Faghri, M. Zinculose: A new fibrous material with embedded zinc particles. Eng. Sci. Technol. Int. J. 2021, 24, 571–578.

[2] Charbaji, A.; Heidari-Bafroui, H.; Rahmani, N.; Anagnostopoulos, C.; Faghri, M. A 3D Printed Lightbox for Enhancing Nitrate Detection in the Field Using Microfluidic Paper-Based Devices, Innovations in Microfluidics and Single Cell Analysis, Boston, 2022.  

[3] Kim, S.C.; Jalal, U.M.; Im, S.B.; Ko, S.; Shim, J.S. A smartphone-based optical platform for colorimetric analysis of microfluidic device. Sens. Actuators B Chem 2017, 239, 52–59.

[4] Wang, J.; Yang, L.; Wang, H.; Wang, L. Application of Microfluidic Chips in the Detection of Airborne Microorganisms. Micromachines 2022, 13, 1576.

Finally, while the article is interesting to the readers of Micromachines, I recommend that the author makes the article easier to follow by the readers. The author is encouraged to include much more details and information in the different sections to provide a deeper understanding to the readers. The author is also required to create new sections and sub-sections and to add more figures in text. 

Author Response

Dear respected reviewer and editor,

Thank you!

Reviewer 2 Report

This paper provides a detailed review of LFIA and its application on infectious diseases detection. Below are some minor comments:

1. In 'Abundant immunological essays currently exist for detecting pathogens and identifying infected individuals', essays should be assays?

2. In 'Therefore, developing a detection method that can overcome all these limitations is essential, such as Lateral flow immunoassays (LFIA)', Lateral should be lateral?

3. In 'Described in further detail below are the strip components mentioned above', detail should be 'details'?

4. For the statement 'NCM with a small pore size is associated with a high capillary flow rate and slower membrane', this sentense is confusing, high flow rate and slower at the same time? The author is suggested to add more detailed information.

5. For the statement 'Accordingly, faster running membranes are more favourable for the strips.', I think the flow rate is related to the chemical reaction (such as bonding kinetics). It is not a good idea to simply say faster is better.

6. The item 'Phosphate Buffered Saline' should be phosphate buffered saline?

7. For the statement 'This type of reporter is the fluidic form of gold', maybe more explaintion is needed about the fluidic form..

8. For the 'Physiochemically, these tags resemble simple colloidal gold nanoparticles (AuNP).' Physiochemically should be physicochemically?

9. For the 'a layer of 4 nm AuNP to generate strong colorimetric shill,', shill should be shell?

10. For the 'As COVOD-19 spreads and mutates, it is essential to validate and evaluate LFIA devices to detect SARS-CoV-2 antigens.', COVOD should be Covid?

11. For the substrate of LFIA, now there are some new developed substrates, such as polymer pillar forests and other porous media, below are several examples:

"Silane–dextran chemistry on lateral flow polymer chips for immunoassays", Lab on a Chip, 2008.

"Synthetic Paper Separates Plasma from Whole Blood with Low Protein Loss", Analytical Chemistry, 2020.

12. The author is suggested to add a conclusion to this review.

Author Response

(The authors gave the same response as above.)

Round 2

Reviewer 1 Report

The author did an excellent job in improving the manuscript according to comments and recommendations provided earlier. The manuscript is now suitable for publication in Micromachines and will be of great interest to its readers; however, the author is required to take note of the following:

Proofread their manuscript again to eliminate a few typos present (e.g. “is an anther important”, “coulometric signal”, “market contain(s) surfactant”, “organic fluorescent dye(s)”, “to include hepatitis (instead of including)”, “mesquite”, etc.)

I believe the author meant to say “time” and not “rate” after the second “flow” in “NCM with a large pore size provides lower capillary flow time, while NCM with a small pore size is associated with a high capillary flow rate and slower membrane [29].” This has the opposite meaning and the correct term to use here is “time” i.e. “high capillary flow time and slower membrane” or slower flow rate.

The author is encouraged to keep their initial statement “The dispensing rate is usually between 0.5 to 1 μl/cm to obtain a line width of 1 mm.” However, they have to mention that this is based on their experience in developing LFIAs with certain types of antibodies. 

The author is encouraged to include the term “hook effect” after “high-dose-effect” since that’s what it’s commonly referred to in the field of paper-based and lateral flow devices and there are several ways and techniques to mitigate it.